# Lymph Node Involvement in Recurrent Serous Borderline Ovarian Tumors: Current Evidence, Controversies, and a Review of the Literature

**DOI:** 10.3390/cancers15030890

**Published:** 2023-01-31

**Authors:** Laureline Wetterwald, Apostolos Sarivalasis, Aikaterini Liapi, Patrice Mathevet, Chahin Achtari

**Affiliations:** 1Oncology Department, Lausanne University Hospital, University of Lausanne, 1011 Lausanne, Switzerland; 2Department of Obstetrics and Gynecology, Lausanne University Hospital, University of Lausanne, 1011 Lausanne, Switzerland

**Keywords:** borderline ovarian tumor, low malignant potential ovarian tumor, serous borderline ovarian tumor, lymph node, recurrence, prognosis, hormonal treatment

## Abstract

**Simple Summary:**

Serous borderline ovarian tumors (SBOTs) are typically associated with an excellent prognosis. Recurrences occur in 5–10% of cases, often several years after initial diagnosis, and might display malignant transformation. Lymph node involvement can be found in up to 30% of primary SBOTs, but its prognostic value is controversial. In this review, we discuss the clinical implications of lymph node involvement in recurrent disease.

**Abstract:**

Borderline ovarian tumors (BOTs) account for 10–20% of epithelial ovarian neoplasms. They are characterized by their lack of destructive stromal invasion. In comparison to invasive ovarian cancers, BOTs occur in younger patients and have better outcome. Serous borderline ovarian tumor (SBOT) represents the most common subtype of BOT. Complete surgical staging is the current standard management but fertility-sparing surgery is an option for SBOT patients who are at reproductive age. While most cases of SBOTs have an indolent course with favorable prognosis, late recurrence and malignant transformation can occur, usually in the form of low-grade serous carcinoma (LGSC). Thus, assessment of the recurrence risk is essential for the management of those patients. SBOTs can be associated with lymph node involvement (LNI) in up to 30% of patients who undergo lymph node dissection at diagnosis, and whether LNI affects prognosis is controversial. The present review suggests that recurrent SBOTs with LNI have poorer oncological outcomes and highlights the biases due to the scarcity of reports in the literature. Preventing SBOTs from recurring and becoming invasive overtime and a more profound understanding of the underlying mechanisms at play are necessary.

## 1. Introduction

Borderline ovarian tumors (BOT) were first described by Taylor in 1929 as tumors displaying an intermediate behavior between benign and malignant ovarian tumor [1]. BOTs were acknowledged by the International Federation of Gynecology and Obstetrics (FIGO) in 1971 as tumors of “low malignant potential” [2], and subsequently accepted by the World Health Organization (WHO) in 1973 [3].

BOTs represent 10–20% of all epithelial ovarian tumors [4]. In contrast to invasive ovarian carcinomas, they are diagnosed at early stages, with 75% of patients presenting a stage I disease according to the FIGO staging system, in a younger population, and have better overall prognosis [5].

Histologically, BOTs are characterized by epithelial proliferation and nuclear atypia, but lack destructive stromal invasion [6]. BOTs are classified into six histologic subtypes based on the epithelial cell type, similar to invasive carcinomas. The most frequent subtypes are the serous borderline ovarian tumors (SBOT), representing 50% of all BOTs, and mucinous borderline ovarian tumors (MBOTs), accounting for an additional 45% of all BOTs [7]. SBOTs are associated with microinvasive sites and extra-ovarian spread in the form of non-invasive peritoneal implants and lymph node involvement (LNI) in 20–40% of cases [7,8,9]. Since the 2014 WHO classification, the previously called “invasive implants” are now considered as low-grade serous ovarian carcinoma (LGSOC) [10]. Following the 2020 WHO classification, SBOTs are classified into two subtypes: conventional and micropapillary/cribriform subtypes [11]. At the genetic level, SBOTs display similar alterations as LGSOC, with KRAS and BRAF mutations being detected in about 30% of SBOTs, usually in a mutually exclusive pattern [12].

Globally, BOTs have a favorable prognosis with a 10-year survival of 97% for all stages of combined tumors [5]. However, the 20-year survival rate of patients with SBOTs drops to 76–89% depicting the potential of late recurrence [13,14]. Indeed, 5–10% of the tumors recur, out of which, 20% display malignant transformation, usually in the form of LGSOC [15,16]. Such recurrences are generally found in the pelvis or abdomen [13,17]. Several clinical features predicting recurrence have been suggested, the most consistent being the FIGO stage, micropapillary subtype, and fertility-sparing surgery (FSS), while the impact of LNI is highly debated [18,19,20,21,22,23]. Predictive tools such as the nomogram of Bendifallah and the score of Ouldamer can be useful in assessing the risk of recurrence [24,25]. At the molecular level, *BRAF* mutations are more frequent in SBOTs that did not recur and are suggested to play a protective role against malignant transformation [26,27,28]. In contrast, SBOTs harboring the *KRAS* G12v mutation seems to display a more aggressive behavior with frequent LGSOC transformation in recurrent diseases [29].

According to the FIGO guidelines, complete surgical staging of epithelial ovarian tumors comprises hysterectomy, bilateral salpingo-oophorectomy, omentectomy, peritoneal washing with cytology, resection of peritoneal lesions, systematic peritoneal biopsies, and paraaortic lymphadenectomy [30,31]. However, following this procedure might result in the overtreating of some BOT patients. With one third of affected woman being diagnosed before the age of 40 years [4], FSS with preservation of the uterus and at least one ovary is now considered as an acceptable therapeutic alternative according to the National Comprehensive Cancer Network (NCCN) guidelines [32]. It is, nevertheless, associated with a higher rate of recurrence [33]. However, the risk of recurrence is considered acceptable as most of the recurrences can be salvaged surgically and does not impact the overall survival [34,35,36]. Systematic lymph node (LN) dissection is not part of the standard surgical procedure as the recurrence and survival rate for patients with affected or not affected LNs have been reported to be similar [37,38,39]. To date, there is no evidence that adjuvant therapy decreases the relapse rate and thus is not indicated [40]. Gynecologists perform follow-up of SBOTs without routine imaging studies while recurrent SBOTs, especially in the case of suspicion of invasion, are pre-operatively staged by imaging including CT-scan and MRI.

In this review, we summarized the current knowledge of LNI in recurrent SBOTs and the controversies related to their management of the disease.

## 2. Materials and Methods

The design of the current manuscript followed the recent recommendations on the quality assessment of narrative review articles [41]. We searched for articles reporting cases of primary SBOT that presented with LNI at recurrence. We performed a Medline (Medscape, New York, NY, USA) search incorporating the terms “serous borderline ovarian tumor”, “serous ovarian tumor of low malignant potential”, “recurrence”, and “lymph node”. Only publications in English were considered. No limitations regarding publishing date or article type were applied. The search was conducted between June and August 2022. We identified additional papers by checking the references of the reviews, in addition to references cited in the relevant articles already selected. Eighteen papers were identified. We excluded five papers that were lacking the precise histologic diagnosis of the primary tumor or of the relapse.

## 3. Results

We identified 13 articles reporting in total 26 cases of patients with initial SBOT who presented LNI at recurrence. Table 1 summarizes the clinical features of these cases.

The age at initial diagnosis ranged from 20- to 73-year-olds (mean 40.6-year-old). The initial stage was FIGO I in nine out of 22 patients (4 unknown). The time between primary tumor diagnosis and disease recurrence ranged from 4 months to 25 years (mean 10.4 years). In 18 patients out 25, the recurrence affected extranodal sites in addition to nodal involvement (one unknown).

A summary of the histologic features of the lymph node recurrences, their management, and outcome, is provided in Table 2. Interestingly, in 13 out of 26 cases, the recurring tumor displayed malignant transformation. Malignant transformation at recurrence seemed to worsen the patient prognosis. Among the 13 patients with invasive malignant transformation (12 LGSOC and one HGSOC), five died of the disease. At the last follow-up, ranging from 1 to 17 years after relapse (mean 8.6 years), four were alive with persistent evidence of disease, one of whom had progressive disease. Only three patients had no evidence of disease. The outcome of one patient was unknown. On the other hand, out of the 13 patients with non-invasive LN recurrence, six had no evidence of disease at the last follow-up and only one died of the disease. The outcome of six patients in this group was not reported.

At relapse, among the patients with invasive malignant transformation, nine received chemotherapy, two of whom also underwent salvage surgery, and one patient was treated with tamoxifen. The subsequent treatment was unknown for three patients in this group. Out of the 13 patients with non-invasive LN recurrence, three underwent salvage surgery, one of whom also received adjuvant chemotherapy, and one was treated only with chemotherapy. Treatment decision was lacking among the nine patients in this group.

## 4. Discussion

### 4.1. Frequency and Distribution

The precise incidence of LNI in SBOT is difficult to assess as systematic LN dissection is not considered as a standard procedure, and thus a significant subset of patients was not formally staged. The same applies to pre-operative imaging assessment, subject to significant discrepancies among institutions. Furthermore, the reported incidence of LNI also greatly depends on the extent of the sampling. In two retrospective studies, 20% to 30% of affected women, with a SBOT apparently confined to the ovary, had positive nodes [52,53]. Interestingly, Djordjevic and Malpica reported that younger patients were more at risk of presenting with LNI at diagnosis [43].

Pelvic, mesenteric, and paraaortic LN groups were most commonly involved [9]. However, since only paraaortic/pelvic lymphadenectomy and omentectomy are part of the standard staging procedure for SBOT, it is unclear if other node groups that are not routinely evaluated could be involved. Indeed, supradiaphragmatic node involvement including supraclavicular, mammary, axillary, and cervical regions have been reported in this setting [9,42,44,46,48,49,54,55,56]. Interestingly, supradiaphragmatic involvement was mainly described in cases of disease recurrence. These data point to the limitations and potential report bias of LNI among both the primary and recurrent SBOT clinical settings with a risk of underreporting them.

### 4.2. Pathologic Features

Beyond the detection and reporting of uncertainties of LNI in SBOT, another challenging aspect is the need to distinguish LNI from hyperplastic mesothelial cells and benign glandular inclusions (Müllerian inclusion cysts or endosalpingiosis) by the pathologist [57,58].

In their article, McKenney et al. described four histologic patterns of nodal involvement in BOTs, with multiple patterns often present simultaneously in an individual patient as well as in the individual LN [9]. The most frequent pattern, identified in 90% of patients, is characterized by clusters of cells, individual cells, or simple non-branching papillae within the nodal sinus or parenchyma. The second pattern, designated “intraglandular” is characterized by complex intraglandular papillary proliferation showing secondary and tertiary branching. The third pattern is characterized by aggregates of epithelial cells with prominent eosinophilic cytoplasm. The least common pattern is characterized by prominent micropapillary architecture. These different patterns can spread in the LN sinuses or parenchyma.

They further reported that 45% of the BOT patients had diffuse LNI, defined as any of the previously described morphologic patterns present throughout the LN, but with intervening lymphoid tissue between the clusters of epithelium. Some patients also displayed a nodular aggregate characterized by an epithelial infiltrate arranged in confluent sheets or separated only by desmoplastic stroma, without intervening lymphoid tissue, measuring greater than 1 mm in linear dimension. Interestingly, this feature was more commonly observed in recurrent disease with delayed LNI rather than with LNI present at initial presentation. Tan et al. also observed a different pattern whether the LNI was detected at the initial presentation or at relapse, the first being focal and intrasinusoidal and the latter being massive and intraparenchymal [44]. This report summarizes the patterns of LNI in BOT, especially in the primary setting and offers thoughtful insights into the patterns of LNI that could be observed in the recurrent setting. The articles used in this review did not report on the LN pattern of spread, nor described the molecular alterations present in the SBOT cases to draw further conclusions on the pathogenesis.

### 4.3. Pathogenesis

The association of SBOT with LNI is the subject of debate, especially in the absence of invasive component. Two leading theories, the endosalpingiosis and metastatic theories, summarize the models of LNI in SBOT [59].

In the endosalpingiosis theory, nodal SBOT develops from nodal endosalpingiotic glands, which are benign ectopic inclusions of the Müllerian type, and is independent from the associated ovarian tumor. This ectopic tissue would be susceptible to the same hormonal and genetic alterations as its native counterparts. In support of this theory, cases of morphologic transitions between endosalpingiotic deposits and BOT within the same node have been reported [60,61]. Furthermore, the coexistence of endosalpingiotic glands have been observed in 58% to 63% of patients with nodal involvement [9,62]. Alverez et al. reported identical KRAS mutations in SBOT and adjacent benign Müllerian inclusions, further supporting the endosalpingiosis theory [63]. However, some authors suggest that these Müllerian inclusions are in fact bland-appearing forms of metastatic SBOT [57]. A major argument against the endosalpingiosis hypothesis is the scarcity of reported primary nodal borderline tumors. However, one could argue the fact that these nodes are generally not sampled in the absence of suspected neoplasia.

The metastatic theory suggests that nodal lesions originate from the primary ovarian SBOT. To explain how the noninvasive SBOTs can metastasize, a hypothesis is that microinvasion, which would provide access to lymphatics, is more prevalent than it is currently recognized. However, several studies have found a remarkably low incidence of nodal involvement in SBOTs with microinvasion [48,64,65,66,67]. Other authors have suggested that tumor cells are transported to the LN. After detaching from the ovarian surface through the mechanical breakages of tumoral papillae and entering the peritoneal cavity, the cells could access the lymphatic system and enter the LN sinus without parenchymal destruction. In support of this theory, Camatte et al. found that all patients with LNI had concomitant peritoneal implants [62].

These two putative mechanisms are not mutually exclusive and could coexist in the same individual. In order to determine if the nodal lesions are clonally related to the primary tumors, Emerson et al. studied the pattern of X chromosome inactivation in both locations. They found that the inactivation patterns were identical between the ovarian and nodal lesions in eight out of nine patients. This supports that the majority of LNI develop through a metastatic mechanism [68]. While these theories can explain the LNI in the primary setting, the mechanisms of development of LNI in the recurrent setting nevertheless remain uncertain.

### 4.4. Clinical Significance

In the case of invasive ovarian tumors, pelvic and paraaortic LN dissection is part of the standard procedure aiming at complete surgical staging for early stages and at cytoreduction in advanced disease. The role of lymphadenectomy in SBOT remains debated and the prognostic significance of LNI is currently uncertain.

In order for lymphadenectomy to represent a valuable diagnostic and therapeutic procedure, it should include the dissection of pelvic and paraaortic nodes up to the renal vessels as the rate of abdominal LN is not negligible. The benefits of such a procedure should outbalance its risks. This implies that lymphadenectomy should provide information that affects prognosis and helps determine the need for adjuvant treatment, tailors follow-up, or that it should reduce the risk of recurrence. Currently, in both primary and recurrent SBOTs, the setting data are missing. Our report has clearly highlighted the scarcity of available data in this setting to draw definitive conclusions.

While complete pelvic LN staging has been shown to upstage 24% to 47% of patients with BOT [69], the impact of this procedure on the oncological outcome and SBOT recurrence rates is uncertain. Leake et al. reported a statistically significant increased incidence of recurrence in patients with nodal involvement [52]. Similarly, Ureyen et al. reported that the presence of positive lymph nodes was significantly associated with worse progression-free survival in patients with SBOT [19]. However, other authors could not reproduce these observations [35,38,70]. Similarly, no consistent effect of LNI on survival has been demonstrated [9,35,38,43]. However, McKenney et al. reported a decrease in disease-free survival for patients with LNI containing nodular aggregates and stromal desmoplastic reaction [9].

LNs may be the site of relapse of SBOT and the transformation site to invasive carcinoma, notably LGSOC. Longacre et al. reported that 10% of carcinoma transformations occurred in lymph nodes [35]. Furthermore, rarely, LN dissection may be a diagnostic procedure revealing synchronous LGSOC in both the primary and recurrent setting. As our review has highlighted, half (12/26) of the patients with SBOT and LN involvement in the recurrent setting had LGSOC with one patient suffering a HGSOC. Once invasive ovarian carcinoma is detected, the prognosis, treatment recommendations, and follow-up are significantly altered. Caution is advised though, for the interpretation of these findings due to the risk of bias. Furthermore, although a possible association between LNI and a higher risk of transformation to an invasive LGSOC in the recurrent setting is possible, the lack of systematic surgical LN staging in the primary SBOT treatment renders intricate any further associations.

Taken together, these observations as well as the higher morbidity of LN resection, lead to the currently broadly accepted eviction of systematic LN resection at the first line surgical resection and staging of SBOT. Furthermore, the role of extensive surgical LN staging remains controversial, even in the recurrent and metastatic SBOT setting, being performed once enlarged LNs are detected pre-operatively on imaging.

### 4.5. Adjuvant Therapy and Follow-Up

Currently, adjuvant, post-operative, treatment is not regarded as the standard of care for SBOT, irrespective of their risk factors and the clinical setting. The uncertain evolution of SBOT toward invasive ovarian cancer and the limited efficacy data of adjuvant treatments including chemotherapy and radiotherapy are the main reasons. Trope et al. reviewed four prospective randomized trials of patients with early stage BOT and reported that adjuvant treatment with chemotherapy or radiotherapy did not improve overall survival [40]. Likewise, there is no proven benefit of adjuvant chemotherapy in advanced-stage disease with residual disease or with LNI [71,72].

SBOTs are hormone-sensitive harboring estrogen-receptors (ER) and progesterone-receptors (PR) in the majority of cases (60–90% and 80%, respectively), thus targeting hormones constitute a sound therapeutic target [73,74]. In recent years, compelling evidence suggests a benefit for anti-hormonal treatment in recurrent and metastatic SBOT cases [73,75,76,77,78,79,80,81]. The clinical benefit of anastrozole, an aromatase inhibitor, was recently reported by the phase II, PARAGON trial reported by Tang et al. In this study, anastrozole was associated with a significant clinical benefit among 61% of patients with recurrent ER and/or PR-positive SBOTs [82].

This treatment option could be discussed for high risk SBOTs including the presence of LN metastasis or micro-invasion and impossible or uncomplete surgical resection without any further possibility of subsequent complete debulking as well as in LGSOC. The most effective antihormonal treatment and its duration have yet to be determined. Despite these compelling data, systemic adjuvant treatment for borderline serous ovarian cancer is not a standard treatment. The lack of consensus about the most effective antihormonal treatment and its duration, together with the uncertain characterization of the risk factors affecting the clinical outcome of these tumors, especially in the recurrent setting, makes patient selection uncertain and impacts the treatment decision making. This issue is especially important for young patients with a desire for fertility preservation and more broadly in patients with underlying conditions potentially affected by anti-hormonal treatments.

MEK inhibitors could represent a promising treatment strategy for SBOTs. Similarly to LGSOC, SBOTs have a high frequency of KRAS and BRAF mutations [12]. Clinical trials are still lacking to determine if RAS/RAF/MEK (MAPK pathway) targeted treatments could be of use for the treatment of high-risk SBOTs.

Regular and long-term follow-up is essential for the early detection of SBOT recurrence. While the optimal timing and optimal modalities for SBOT follow-up are still a matter of debate, several articles have supported the use of transvaginal ultrasound in combination with systemic imaging, physical examination, and tumor-marker dosing for recurrence monitoring [83,84,85].

## 5. Conclusions

Maximal surgical debulking can successively treat primary or recurrent SBOT. The presence of lymph node metastasis without concomitant invasive malignant tumor, is often disturbing for oncologists and is challenging for surgical planning, especially in the recurrence setting. Despite the controversies present in the literature, this review suggests that LN metastasis affects oncological outcome in recurrent SBOT. It has been reported that 20% of SBOTs present invasive malignant transformation at recurrence, though our review reported a 50% invasive malignant transformation rate (13/26), and associated poor prognosis with a 38% of mortality rate (5/13). While these results are limited by potential report biases and surgical under-diagnosis, these results are significant and require further study. Only by preventing SBOT recurrence and invasive transformation, and with a better understanding of the underlying molecular mechanisms, can the outcomes of this disease be improved in the future.

## Figures and Tables

**Table 1 cancers-15-00890-t001:** Clinical characteristics of cases of SBOT with lymph node involvement at recurrence.

Author	Year of Publication	Patient, (*n*)	Age (year)	FIGO Stage	Interval (year)	Extranodal Involvement	Lymph Node
Zanetta et al. [34]	2001	1	26	IIB	4	Yes	Obturator
Moreira et al. [42]	2002	1	28	N/A	1	N/A	Intramammary
Djordjevic et al. [43]	2010	1	N/A	N/A	10	No	Para-aortic, perirenal, supraclavicular
		1	N/A	N/A	9	No	Supraclavicular
Longacre et al. [35]	2005	1	56	IA	25	Yes	Axillary
		1	33	II	19	Yes	Axillary
		4	N/A	II–IV	N/A	Yes	Pelvic, para-aortic, axillary, supraclavicular
		1	N/A	I	N/A	Yes	Axillary, supraclavicular
Tan et al. [44]	1994	1	50	III	4	Yes	Omental
		1	73	III	5	Yes	Scalene
		1	27	III	7	Yes	Cervical
Quilichini et al. [45]	2020	1	25	IIIA	6	Yes	Para-aortic
Chamberlin et al. [46]	2001	1	66	III	7	No	Internal mammary
Parker et al. [47]	2004	1	50	IC	2.4	Yes	Axillary
Prat et al. [48]	2002	1	N/A	I	4	Yes	Supraclavicular
Malpica et al. [49]	2001	1	29	IIA	0.3	No	Axillary, intramammary
		1	43	IC	20	Yes	Scalene
		1	27	IB	14	No	Neck
Alagkiozidis et al. [50]	2012	1	20	N/A	25	No	Supraclavicular, retroperitoneal, paracaval, celiac, paratracheal, subcarinal
Silva et al. [17]	1998	1	43	I	20	Yes	Neck
		1	41	I	7	Yes	Neck
		1	33	I	18	No	Pleural
Michael et al. [51]	1985	1	50	III	10	Yes	N/A

FIGO, International Federation of Gynecology and Obstetrics; N/A, not available.

**Table 2 cancers-15-00890-t002:** Pathology, management, and outcome of primary SBOT with nodal involvement at recurrence.

Author	Histologic Features of Recurrent Disease	Treatment	Follow-Up
Zanetta et al. [34]	Malignant transformation (LGSOC)	Surgery + CHT	NED
Moreira et al. [42]	SBOT	N/A	N/A
Djordjevic et al. [43]	Malignant transformation (LGSOC)	CHT	AWD
	Malignant transformation (LGSOC)	CHT	AWD
Longacre et al. [35]	Malignant transformation (LGSOC)	Tamoxifen	NED
	Malignant transformation (LGSOC)	CHT	DOD
	SBOT	N/A	N/A
	SBOT	N/A	N/A
Tan et al. [44]	SBOT	N/A	NED
	SBOT	N/A	NED
	SBOT	N/A	NED
Quilichini et al. [45]	SBOT	Surgery	NED
Chamberlin et al. [46]	SBOT	Surgery	NED
Parker et al. [47]	Malignant transformation (HGSOC)	CHT	DOD
Malpica et al. [49]	SBOT	CHT	NED
	Malignant transformation (LGSOC)	CHT	DOD
	Malignant transformation (LGSOC)	Surgery + CHT	NED
Prat et al. [48]	SBOT	Surgery + CHT	DOD
Alagkiozidis et al. [50]	Malignant transformation (LGSOC)	CHT	N/A
Silva et al. [17]	Malignant transformation (LGSOC)	N/A	DOD
	Malignant transformation (LGSOC)	N/A	AWPD
	Malignant transformation (LGSOC)	N/A	DOD
Michael et al. [51]	Malignant transformation (LGSOC)	CHT	AWD

SBOT, serous borderline ovarian tumor; LGSOC, low-grade serous ovarian carcinoma; HGSOC, high-grade serous ovarian carcinoma; CHT, chemotherapy; DOD, dead of disease; AWD, alive with disease; AWPD, alive with progressive disease; NED, no evidence of disease.

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
