# Peer review of "Lymph Node Involvement in Recurrent Serous Borderline Ovarian Tumors: Current Evidence, Controversies, and a Review of the Literature"

_cancers, 2023, doi:10.3390/cancers15030890_

Round 1
Reviewer 1 Report
This article is an interesting review related to a rare condition. In fact, the prognostic impact of lymph node coninvolvement of borderline tumors is not known. The article is well structured and well written. The literature review is comprehensive and well reported. I suggest that the discussion regarding the literature review should be made clearer. In particular, it is not clear whether the lymph node involvement of the cases reported in the literature is from a borderline tumor or from a low-grade serous carcinoma. In the latter case, it would be more correct to use the word: Carcinoma and not BOT. I would also place emphasis on the different outcomes of a lymph node involvement from BOT versus a lymph node metastasis from carcinoma (evolved from a BOT) because this changes in clinical and pathological terms.
I also suggest that the authors refer to the most recent classification of tumors: WHO Classification of Tumours, Female Genital Tumours, 5th Edition, 2020 and to cite some review articles of that classification: DOI: 10.3390/diagnostics11040697 Also I would like you to pay attention to this recent very similar article doi: 10.1016/j.ygyno.2021.05.033. Plagiarism should be avoided.
Author Response
Point 1: I suggest that the discussion regarding the literature review should be made clearer. In particular, it is not clear whether the lymph node involvement of the cases reported in the literature is from a borderline tumor or from a low-grade serous carcinoma. In the latter case, it would be more correct to use the word: Carcinoma and not BOT. I would also place emphasis on the different outcomes of a lymph node involvement from BOT versus a lymph node metastasis from carcinoma (evolved from a BOT) because this changes in clinical and pathological terms.
Response 1: We thank the reviewer for his suggestions and comments. We provided more clarity in the discussion especially clarifying that the whole paper was about serous BOT (SBOT), a precursor of low-grade serous ovarian cancer (LGSOC). We made the discussion clearer to state the association of lymph-node involvement (LNI) with SBOT. Interestingly somehow, and this is the point of the paper in recurrent SBOT not all lymph-node relapses are associated with invasive LGSOC. An emphasis was added in this respect to the manuscript discussion.
Point 2: I also suggest that the authors refer to the most recent classification of tumors: WHO Classification of Tumours, Female Genital Tumours, 5th Edition, 2020 and to cite some review articles of that classification: DOI: 10.3390/diagnostics11040697
Response 2: We thank the reviewer for his suggestion. we added in the manuscript the updated classification.
Point 3: Also I would like you to pay attention to this recent very similar article doi: 10.1016/j.ygyno.2021.05.033. Plagiarism should be avoided.
Response 3: We thank the reviewer for pointing out this interesting article. The current manuscript is a genuine work without plagiarism so ever. Furthermore, the scope of the paper is different, as we focus on the SBOT recurrent setting and the presence of LNI.

Reviewer 2 Report
Thank you for giving me an opportunity to review the interesting manuscript entitled: Lymph node involvement in recurrent serous borderline ovarian tumors: current evidence, controversies, and a review of the literature.
The topic of the review is interesting enough to gain readers attention. The manuscript is well written and should be accepted after minor revision nevertheless some issues should be noticed.
What is the clinical significance of the molecular profile of the BOTs? Should we predict more aggressive course of the disease due to specific genetic mutations? This topic should be discussed in more extensive manner in the discussion part of the manuscript.
What kind of follow up was implemented in the reviewed publications? How the recurrent LNI was diagnosed? What is the preferable diagnostic tool in such group of patients after primary surgery?
References should be checked and corrected according to Cancers recommendations.
Author Response
Point 1: What is the clinical significance of the molecular profile of the BOTs? Should we predict more aggressive course of the disease due to specific genetic mutations? This topic should be discussed in more extensive manner in the discussion part of the manuscript.
Response 1: We thank the reviewer for his suggestions. This is a very interesting topic and we included in the introduction and the pathogenesis section of our discussion a summary of the available published data. The current evidence is too limited to extract definitive conclusions about the association of molecular profile of SBOT and the risk of LN recurrence. The data on MAPK alterations are definitely more solid for LGSOC. We have included this limitation in our manuscript. Further studies are needed in this field.
Point 2: What kind of follow up was implemented in the reviewed publications? How the recurrent LNI was diagnosed?
Response 2: We thank the reviewer for his comment: This information is lacking in the majority of the reported cases with recurrent SBOT with LNI.
Point 3:What is the preferable diagnostic tool in such group of patients after primary surgery?
Response 3: We thank the reviewer for his query. This field is still a matter of debate among the community and was included in our discussion. Gynecologist performs the follow-up of non-invasive SBOT often by a delayed laparoscopy re-stadification and CA 125 monitoring. Imaging is not routinely performed. In the case of recurrent SBOT with LNI one could argue to implement a follow up as per invasive LGSOC.
Point 4: References should be checked and corrected according to Cancers recommendations
Response 4: We thank the reviewer for his suggestion: we checked and adapted the references according to Cancers recommendations.
Reviewer 3 Report
The authors reviewed the clinical significance of Serous Borderline Ovarian Tumor (SBOT), which is considered a precursor lesion to Low-grade Serous Ovarian Cancer (LGSOC), especially lymph node metastasis. Cases with lymph node metastasis at the time of recurrence were reviewed and fully discussed. In borderline malignancies, lymph node dissection has been reported to have no prognostic impact, and lymph node dissection is not included in the standard procedure. On the other hand, as the authors point out, the incidence of lymph node metastasis is relatively high in serous borderline malignancies. As with peritoneal and omental invasive implants, the presence of lymph node involvement may cause progression to carcinoma or recurrence, potentially affecting the prognosis. As a Reviewer, I have a few questions.
1. Table 1 and Table 2 are listed with a focus on recurrent BOT. However, the discussion is mainly about lymph node metastasis of SBOT, which is a discrepancy from the title. It is not clear whether the authors want to discuss Serous BOT or Total BOT. Also, the discussion seems to lack a comparison between LGSOC and SBOT.
2. References are through 2020, with no references beyond 2021.
3. P2L44: Currently, the 5th edition of the WHO classification of ovarian tumors will be published in 2020. The most recent version should be referred to. Reference #10
4. Material and Methods: Why is "serous" not included in the search terms? The date of the search should be clearly stated. Authors should also specify the number of papers found in the search and the number of papers excluded.
5. Results: In Table 1, most of the references are single case reports and include older references; in addition to the name of the Author, the year of the report of the reference should be given. It may be necessary to consider the surgical procedure, especially whether fertility preservation surgery or tumor enucleation is being performed.
6. We think the Conclusion is too long. Part of it should be a summary of the discussion.
7. Would you be able to refer to the following previous report?
Quin XQ, et al. Clinical Predictors of Recurrence and Prognostic Value of Lymph Node Involvement in the Serous Borderline Ovarian Tumor. Int J Gynecol Cancer. 2018 Feb;28(2):279-284.
Ureyen I, et al. The Factors Predicting Recurrence in Patients With Serous Borderline Ovarian Tumor. Int J Gynecol Cancer. 2016 Jan;26(1):66-72.
Author Response
Point 1: Table 1 and Table 2 are listed with a focus on recurrent BOT. However, the discussion is mainly about lymph node metastasis of SBOT, which is a discrepancy from the title. It is not clear whether the authors want to discuss Serous BOT or Total BOT. Also, the discussion seems to lack a comparison between LGSOC and SBOT.
Response 1: We thank the reviewer for his comments. The current paper focuses on SBOT with LNI in the recurrent setting as the title suggests. We have reviewed the manuscript to make it cleared and correct any potential discrepancies. Both tables were updated excluding report with doubts about the BOT subtype. We also adapted text in order to emphasis on the different outcomes in the case of recurrent SBOT vs LGSOC transformation.
Point 2: References are through 2020, with no references beyond 2021.
Response 2: We thank the reviewer for his comment. In the specific setting covered by this article, all the available up-to-date published data were included. We included in the discussion references beyond 2021.
Point 3: P2L44: Currently, the 5th edition of the WHO classification of ovarian tumors will be published in 2020. The most recent version should be referred to. Reference #10
Response 3: We thank the reviewer for his comment; as with reviewer 1 we included the current WHO stadification. Interestingly though, the majority of published data used the previous classification.
Point 4: Material and Methods: Why is "serous" not included in the search terms? The date of the search should be clearly stated. Authors should also specify the number of papers found in the search and the number of papers excluded.
Response 4: We thank the reviewer for his comments. Indeed, the term “serous” was included in all our search. The text was corrected accordingly. We accordingly added the date of search, number of papers found and number of papers excluded.
Point 5: Results: In Table 1, most of the references are single case reports and include older references; in addition to the name of the Author, the year of the report of the reference should be given. It may be necessary to consider the surgical procedure, especially whether fertility preservation surgery or tumor enucleation is being performed.
Response 5: We thank the reviewer for his comments. As suggested, we included the date of publication. As the type of surgical procedure was not clearly described in all cases, we did not include this information in the table.
Point 6:We think the Conclusion is too long. Part of it should be a summary of the discussion.
Response 6: We thank the reviewer for his comment. The conclusion was reduced in size.
Point 7: Would you be able to refer to the following previous report?
Quin XQ, et al. Clinical Predictors of Recurrence and Prognostic Value of Lymph Node Involvement in the Serous Borderline Ovarian Tumor. Int J Gynecol Cancer. 2018 Feb;28(2):279-284.
Ureyen I, et al. The Factors Predicting Recurrence in Patients With Serous Borderline Ovarian Tumor. Int J Gynecol Cancer. 2016 Jan;26(1):66-72.
Response 7 : we thank the reviewer for his suggestion: we incorporated the above-mentioned papers to our review.
Round 2
Reviewer 3 Report
The authors responded promptly and appropriately to the reviewers' comments. I greatly appreciate your sincere response. The text and tables have been improved to clarify the main concept of the paper.